# HER2 Overexpression and Cytogenetical Patterns in Canine Mammary Carcinomas

**DOI:** 10.3390/vetsci9110583

**Published:** 2022-10-22

**Authors:** L. V. Muscatello, F. Gobbo, E. Di Oto, G. Sarli, R. De Maria, A. De Leo, G. Tallini, B. Brunetti

**Affiliations:** 1Department of Veterinary Medical Sciences, University of Bologna, Ozzano dell’Emilia, 40064 Bologna, Italy; 2OACP IE Ltd., T12H1XY Cork, Ireland; 3Department of Veterinary Sciences, University of Turin, 10095 Grugliasco, Italy; 4Department of Experimental, Diagnostic and Specialty Medicine, University of Bologna, 40138 Bologna, Italy; 5Solid Tumor Molecular Pathology Laboratory, IRCCS Azienda Ospedaliero-Universitaria di Bologna, 40138 Bologna, Italy

**Keywords:** *HER2* gene amplification, canine mammary carcinoma, fluorescence in situ hybridization, tissue microarray, immunohistochemistry, estrogen receptor, Ki67

## Abstract

**Simple Summary:**

Human epidermal growth factor receptor 2 (HER2) is a transmembrane tyrosine kinase receptor whose homonymous gene is an important oncogenic driver in 10–20% of invasive breast cancer. The role of HER2 in canine mammary carcinoma is controversial; therefore, this study aimed to investigate the protein expression and cytogenetic alterations of HER2 and their correlation with the proliferative fraction, estrogen receptor, and other clinicopathological parameters in canine mammary carcinoma. The *HER2* gene was amplified in a subgroup of canine mammary carcinomas, and the *HER2* gene copy number was correlated with HER2 protein overexpression but not with the tumor’s biological behavior. Surprisingly, a possible translocation of *HER2/CRYBA1* was found. This is the first study in the canine species that found cytogenetic alterations with fluorescence in situ hybridization in mammary carcinoma.

**Abstract:**

Human epidermal growth factor receptor 2 (HER2) is a tyrosine kinase receptor that promotes tumor cell growth and is implicated in the pathogenesis of human breast cancer. The role of HER2 in canine mammary carcinomas (CMCs) is not clear. Therefore, this study aimed to examine the protein expression and cytogenetic changes of HER2 and their correlation with other clinical–pathological parameters in CMC. We retrospectively selected 112 CMCs. HER2, ER, and Ki67 were assessed by immunohistochemistry. HER2 antibody validation was investigated by immunoblot on mammary tumor cell lines. Fluorescence in situ hybridization (FISH) was performed with probes for HER2 and CRYBA1 (control gene present on CFA9). HER2 protein overexpression was detected in 15 carcinomas (13.5%). A total of 90 carcinomas were considered technically adequate by FISH, and 8 out of 90 CMC (10%) were *HER2* amplified, 3 of which showed a cluster-type pattern. HER2 overexpression was correlated with an increased number of *HER2* gene copies (*p* = 0.01; R = 0.24) and overall survival (*p* = 0.03), but no correlation with ER, Ki67, grade, metastases, and tumor-specific survival was found. Surprisingly, co-amplification or polysomy was identified in three tumors, characterized by an increased copy number of both *HER2* and *CRYBA1*. A morphological translocation-fusion pattern was recognized in 20 carcinomas (22%), with a co-localized signal of *HER2* and *CRYBA1*. HER2 is not associated with clinical–pathological parameters of increased malignancy in canine mammary tumors, but it is suitable for studying different amplification patterns.

## 1. Introduction

Cancer cells enable genomic instability and acquired recurrent oncogenes amplification and, therefore, replicative advantage, invasion, and metastasis [1].

Human epidermal growth factor receptor 2 (HER2) is a transmembrane tyrosine kinase receptor and member of the human epidermal growth factor receptor family, including HER1, HER3, and HER4, that regulates normal cell proliferation, development, and survival [2]. HER2 has a well-known driver role in the carcinogenesis of breast cancer [3]; it is overexpressed in around 10–20% of breast tumors, and it is associated with aggressive behavior, high recurrence rates, and increased mortality [2,4] in both invasive and in situ ductal carcinoma [5]. Patients with HER2-positive breast cancer are eligible for target therapy [2]. Over the years, several monoclonal antibodies (trastuzumab, pertuzumab, lapatinib, neratinib, and trastuzumab emtansine) have been developed, and their therapeutic efficacy has been tested and validated [4,6].

HER2 is detected in human beings with the combination of protein expression using immunohistochemistry and gene amplification with the gold-standard method of fluorescence in situ hybridization (FISH). In human breast cancer, the HER2 evaluation system and experimental methodology are constantly being updated and revised by a group of expert pathologists and oncologists of the American Society of Clinical Oncology and the College of American Pathologists (ASCO/CAP) [7,8,9] to continuously ameliorate the evaluation of the clinical significance of HER2 assessment. An immunohistochemical score of 3+ is considered positive for receptor overexpression, while tumors that result in 2+ on immunohistochemistry are considered equivocal and further tested with FISH to investigate the *HER2* gene [7]. 

The biological role of HER2 overexpression in canine mammary carcinomas (CMC) is not straightforward, with discordant results in the literature. Different studies have demonstrated that HER2 is not associated with prognostic variables [10,11] or poor prognosis in the canine species [12], with work even showing its association with an increased survival rate [13]. Conversely, other authors have reported that HER2-positive tumors correlate with negative clinicopathological parameters, such as histological grade, nuclear pleomorphism, and mitotic count [14,15].

Other than canine mammary carcinoma [10,11,12,13,14,15,16], immunohistochemical overexpression of the HER2 protein has recently been documented in lung, prostate, urinary bladder, anal sac, thyroid, skin, and gastrointestinal tract carcinomas [17,18,19,20,21,22,23].

In the canine species, HER2 protein expression is widely investigated [10,11,12,13,14,15,16], but little is known about its genetic alterations. 

In canine urothelial carcinoma, HER2 was evaluated with both immunohistochemistry for protein expression and PCR for the copy number aberration status [21]; in another study on canine apocrine anal sac gland carcinomas, the authors demonstrated both HER2 mRNA and protein expression [19].

The pathway between HER2 overexpression and gene amplification was investigated previously in canine mammary tumors using immunohistochemistry and chromogenic in situ hybridization, respectively [24]. No amplification was detected, even in overexpressed cases [24]. Therefore, the aims of the present studies were (1) to analyze the expression of HER2 in canine mammary carcinoma and its correlation with the amplification status; (2) to investigate the correlation of these results with clinicopathological parameters in canine species. Additionally, (3) we aimed to demonstrate the cross-reactivity of the anti-human HER2 antibodies used, considering the discrepancies present in the literature.

## 2. Materials and Methods

### 2.1. Caseload

From the archives of the Anatomic Pathology Service of the Department of Veterinary Medical Sciences, University of Bologna, and the private AniCura Veterinary Hospital I Portoni Rossi of Zola Predosa, we retrospectively collected 112 canine mammary carcinomas. Moreover, we collected 17 benign tissues composed of benign neoplasm (7 simple adenomas and 1 benign mixed tumor), hyperplastic (7 cases), and normal (2 cases) mammary glands used as diploid controls for FISH analysis. The samples were available as formalin-fixed, paraffin-embedded material and routinely stained hematoxylin and eosin (HE) histological sections. 

The anamnestic and clinical data, such as breed, age, sex (all female, 74 intact and 38 spayed), anatomical location, and tumor size, were collected. Histological diagnosis was assessed based on the current classification of canine mammary tumors [25]. The histological grade was evaluated based on tubule formation, nuclear pleomorphism, and mitotic count following the canine-adapted Nottingham system [26]. 

### 2.2. Western Blot for Antibody Validation

Proteins from canine mammary carcinoma cell lines CypP and CypM [27] and human ovarian carcinoma cell line SKOV3 were extracted in lysis buffer (1% Triton X-100, 10% glycerol, 50 mM Tris, 150 mM sodium chloride, 2 mM EDTA, pH 8.0, and 2 mM magnesium chloride) containing protease inhibitor cocktail (P8340, Sigma Aldrich, Darmstadt, Germany). An amount of 20 micrograms of total proteins was separated in SDS-PAGE (10% or 15%) and transferred onto a PVDF membrane. After washing, membranes were incubated in TBS/BSA 10% (bovine serum albumin) at room temperature for 1 h and then incubated overnight at 4 °C with a polyclonal antibody anti-HER2 (A0485, Dako, Glostrup, Denmark). After incubation, membranes were washed in TBS/Tween 0.01%, and specific bands were revealed by horseradish peroxidase secondary antibodies.

### 2.3. Immunohistochemistry 

A total of 112 tumors and 17 control samples were included in the immunohistochemical analysis. The sections were cut 3 microns thick and underwent immunohistochemistry (IHC) using antibodies to HER2 (polyclonal A0485, Dako, Glostrup, Denmark; dilution 1:200), ER (polyclonal, ThermoFisher Scientific, Waltham, Massachusetts, USA; diluition 1:200), and Ki67 (MIB-1, Dako, Glostrup, Denmark; dilution 1:600). The immunohistochemical method was performed as in the previously published works [28,29]. Sections of normal canine mammary gland and uterus for ER, normal gut for Ki67, and canine mammary carcinoma previously scored 3+ for HER2 were used as positive controls to assess the specificity of the immunohistochemical procedure. Primary antibody was replaced with an irrelevant, isotype-matched antibody as a negative control.

The evaluation of the immunohistochemical expression of HER2 was recorded according to the current recommendations of the American Society of Clinical Oncology/College of American Pathologists (ASCO/CAP) [7]. Immunoreactivity to HER2 is divided into the following classes: 3+ (positive), 2+ (equivocal), and 0 and 1+ (negative). The ASCO/CAP algorithm is based on the cellular membranous intensity of staining and the percentage of positive tumor cells (cut-off value of 10%).

ER was evaluated based on the canine-adapted Allred score, with score <3 considered negative [30].

Ki67 index was evaluated by counting the number of Ki67-positive cells per 500 neoplastic cells. Cell count was performed using a digital cell counter (Image J, version 1.52t, National Institutes of Health, Bethesda, MD, USA) [28].

### 2.4. Tissue Microarray 

Tissue microarrays (TMA) were built according to the previous published method [28,29,31]. Twenty double-core TMA blocks were built. Each TMA tumor block included 6 or 7 cases. A total of 2 blocks were composed of benign tissues, including a total of 13 cases (5 adenomas, 1 mixed mammary tumor, 6 mammary hyperplasia, and 1 normal mammary gland), and 18 blocks were composed of 110 malignant tumors. The cores were asymmetrically arranged, and a different tissue (lung and prostate) was inserted as negative control and orientation core.

### 2.5. Fluorescent in Situ Hybridization Experiment Design and Blast Analysis 

To test the possibility of using a human commercial probe, the homology of the nucleotide sequence of *HER2* between the canine species and the human species was verified using the database BLAST (Basic Local Alignment Search Tool—NCBI, Bethesda, MD, USA). The alignment showed an 87% homology of the gene sequence between canine and human species.

Furthermore, to investigate the presence of polysomy, a possible housekeeping gene was searched. The selection criteria were the absence of gene involvement in the tumorigenesis of the human breast tumor or canine mammary tumor, location on chromosome 9 (as *HER2*) in the canine species (CFA9), and high homology of the nucleotide sequence between canine and human species. The *CRYBA1* gene met these criteria; the homology between human and canine species was 92% with BLAST analysis.

### 2.6. Fluorescence in Situ Hybridization Method

A total of 20 TMA sections, 3 microns thick, composed of a total of 110 tumors, and 2 whole sections were subjected to FISH analysis. The FISH method was performed based on a previous publication [29]. Timing, probes, and specification of the present experiment are summarized in Appendix A. Human breast cancer with *HER2* amplified was used as positive control. The specificity of FISH was further evaluated considering the euploidy of the fibroblasts and lymphocytes adjacent to the neoplastic cells. The Image analysis software cytogenetic specific (The CytoVision^®^, Leica Biosystem, Nussloch, Germany) was used for counting the number of gene copies per nucleus in at least 20 tumor nuclei for each case.

The evaluation system used to interpret FISH sections is based on the ASCO/CAP guidelines [7]. Considering that this is the first study in which *HER2* was evaluated with FISH in canine species, both single- (*HER2*) and double-probe (*HER2/CRYBA1*) evaluation systems were applied. The single-probe evaluation algorithm by Wolff et al. [7] takes into account *HER2* gene mean copy number (>6; ≥4 < 6; <4 mean copy number cell/signal) combined with the immunohistochemical expression of HER2 (3+; 2+; 0–1). The dual-probe ISH evaluation system is a complex consequential algorithm that classifies the amplification status of *HER2* based on the combined evaluation of the gene and centromere ratio (*HER2*/CEP17 ≥ 2; *HER2*/CEP17 < 2), the *HER2* gene copy numbers (>6; ≥4 < 6; <4 mean copy number cell/signal), and the immunohistochemical expression of HER2 (3+; 2+; 0–1). To investigate the double-probe system algorithm in canine mammary tumors, we applied the *HER2/CRYBA1* ratio.

Samples were considered indeterminate for *HER2* if technical issues prevented one or both tests (IHC or ISH) from being reported as positive, negative, or equivocal. The conditions of technical errors include artifacts and pre-analytic conditions [8].

### 2.7. Follow-Up Data

A 24-month clinical follow-up was performed. Lymph node metastasis was detected on histological examination. Tumor recurrence, systemic metastases, and survival status were investigated and collected by phone calls to the referent veterinarian or the owner of the dogs. Recurrence is defined as the time elapsed from the first tumor diagnosis and tumor recurrence in the same anatomical site. Tumor-specific survival (TSS) was defined as the time elapsed from the tumor diagnosis to the animal’s death due to the tumor. Tumor overall survival (OS) was defined as the time elapsed from the tumor diagnosis to the animal’s death due to any cause [32,33].

### 2.8. Statistical Analysis

Statistical analysis was performed with the software GraphPad Prism (version 8.3, La Jolla, San Diego, CA, USA). The normality distribution of the data was assessed with the D’Agostino and Pearson Omnibus. Continuous data, not normally distributed, were analyzed with Spearman and Mann–Whitney tests. A log-rank test was used to evaluate the survival variables, and for patients with multiple tumors, the highest-grade tumor was selected. A *p* value ≤ 0.05 was considered significant.

## 3. Results

### 3.1. Caseload, Histological Diagnosis, and Grade

This study was conducted on a retrospective caseload of 92 dogs with 112 primary mammary tumors. The patients were all female, with 55 intact and 37 spayed.

The mean age of the patients at the diagnosis was 10.12 years ± 2.71 (mean and standard deviation). The median tumor size was 19 mm in diameter (range of 2–70 mm).

Tumors were histologically classified as complex carcinoma (36/112), carcinoma mixed type (20/112), solid carcinoma (17/112), simple tubulo-papillary carcinoma (15/112), simple tubular carcinoma (7/112), comedocarcinoma (6/112), intraductal papillary carcinoma (3/112), invasive micropapillary carcinoma (2/112), inflammatory carcinoma (2/112), anaplastic carcinoma (1/112), ductal carcinoma (1/112), lipid-rich carcinoma (1/112), and adenosquamous carcinoma (1/112). Tumors were grade I in 60 cases (54%), grade II in 25 cases (22%), and grade III in 27 cases (24%).

The histological grade was significantly associated with the tumor size (*p* = 0.01, R = 0.372; Spearman test), lymph node (*p* < 0.0001, R = 0.64; Spearman test), and systemic metastases (*p* = 0.004, R = 0.50; Spearman test). The OS was negatively correlated with the tumor size (*p* = 0.01, R = −0392; Spearman test). 

The histological grade was significantly associated with OS (*p* = 0.01 log-rank test) but not with the TSS (*p*= 0.11 log-rank test). DFS data collected were not numerically significant; therefore, they were excluded from the study.

### 3.2. HER2 Expression in Canine Carcinoma Mammary Cell Lines

To validate antibody specificity used by IHC, a Western blot was performed on protein lysate from canine carcinoma cell lines CypP and CypM [27] and human ovarian carcinoma cell line SKOV3. In Figure 1, specific bands corresponding to 185KD HER2 are present in canine cell lines co-migrating with human HER2 over-expressed in the human SKOV3 cell line (positive control). 

### 3.3. HER2 Protein Expression, Ki67 Proliferation Index, and Estrogen Receptors Expression

A total of 15 out of 112 carcinomas (13.5%) showed a complete intense membrane expression of HER2 in > 10% of tumor cells and were considered positive (3+ score). Thirty-seven out of one hundred and twelve carcinomas (33%) were interpreted as equivocal (2+ score) because they showed a complete moderate or scant HER2 expression or basolateral expression. The number of negative cases was 60/112 (53.5%), showing either a weak incomplete membrane expression (1+ score, 45/112 (40%)) or no immunoreactivity (0 score, 15/112 (13.5%)) (Figure 2). No correlation with tumor size, histological grade, Ki67 index, or ER status was found. HER2 protein expression was not associated with OS (*p* = 0.85 log-rank) and TSS (*p* = 0.06 log-rank). HER2 expression in normal mammary gland, lobular hyperplasia, and benign tumors was characterized by an incomplete weak to moderate membrane labeling.

The median Ki67 labeling index in the examined carcinoma was 16% (1–62.8% range). Ki67 correlated with the tumor size (*p* = 0.006, R = 0.37; Spearman test), with presence of lymph node (*p* < 0.0001, R = 0.67; Spearman test) and systemic metastases (*p* = 0.006, R = 0.53; Spearman test). 

The number of ER immunohistochemically positive tumors was 22 out of 91 cases (24%). The negative tumors (score from 0 to 3) were 69/91 cases (76%). In 21 cases, ER expression was not evaluable due to sample exhaustion or lack of reactivity in the adjacent normal mammary gland. ER expression was inversely correlated with the smallest tumor size (*p* = 0.009, R−0.365; Spearman test), lower histological grade (*p* = 0.003, R = −0.31; Spearman test), and a lower Ki67 index (*p* = 0.01, R = −0.28; Spearman test). OS was associated with ER expression (*p* = 0.04, log-rank), and tumor-specific survival cannot be determined due to the absence of tumor-specific death for ER-positive tumors (Figure 3).

### 3.4. HER2 and CRYBA1 Copy Number Variations

Out of 112 mammary carcinomas that underwent FISH analysis, 22 carcinomas were technically inadequate and therefore considered underdetermined. Epithelial cells of the normal mammary glands, lobular hyperplasia, and benign neoplasms showed diploid *HER2* and *CRYBA1* signals. 

With the single-probe evaluation system, considering only the *HER2* gene red signal, 88 adequate samples were amplified in 8 cases (10%) and not amplified in 82 cases (90%). Based on the current ASCO/CAP algorithm, the amplified cases showed three categories of expression:(1)Cases with mean *HER2* copy numbers per nucleus > 6 (4/8 cases).(2)Cases with mean *HER2* copy numbers ≥4 < 6 and with positive immunohistochemical expression (HER2 protein score 3+) (1/8 cases).(3)Cases with mean *HER2* copy numbers ≥4 < 6 and with equivocal immunohistochemical expression (HER2 protein score 2+) that underwent dual-probe evaluation and showed a *HER2/CRYBA1* copy number ratio > 2 (3/8 cases).

With dual-probe evaluation, when considering the combined signals of *HER2* (red signal) and *CRYBA1* (green signal), 69 (90%) tumors were not amplified (Figure 4A), and 8 (10%) tumors were *HER2*-amplified (Figure 4B). Based on the current ASCO/CAP algorithm, the amplified cases showed two categories:(1)Cases with *HER2/CRYBA1* > 2, with a mean of *HER2* copy numbers per nucleus > 4, regardless of the immunohistochemical expression (7/8).(2)Cases with *HER2/CRYBA1* < 2, with a mean of *HER2* copy numbers per nucleus > 4, and with positive immunohistochemical expression (HER2 protein score 3+) (1/8).

**Figure 4 vetsci-09-00583-f004:**
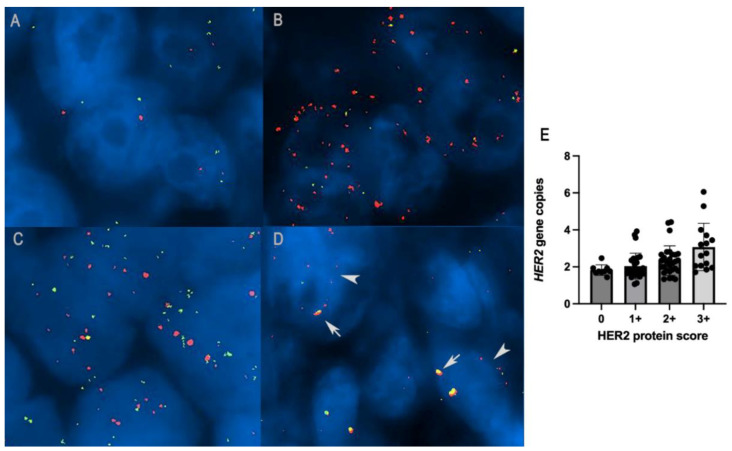
(**A**) *HER2* diploid in canine mammary carcinoma: two gene copies of both *HER2* (red signal) and *CRYBA1* (green signal), FISH. (**B**) *HER2* amplification in canine mammary carcinoma: *HER2/CRYBA1* > 2, FISH. (**C**) *HER2/CRYBA1* co-amplification/polysomy: increased copy numbers of both *HER2* and *CRYBA*1 (> 4 gene copies). (**D**) *HER2/CRYBA1* fusion pattern: close co-localization of *HER2* and *CRYBA1*, interpreted as translocation and fusion of the two genes (arrows) together with non-fused alleles in the same nuclei (arrowheads). (**E**) Histogram representation of the correlation between HER2 overexpression with an increased number of *HER2* gene copies (*p* = 0.01, R = 0.24; Spearman test).

Single-probe and dual-probe evaluation systems produced highly correlated results (*p* < 0.0001, R = 0.88; Spearman test).

Special cytogenetic expression patterns were recognized: *HER2* cluster-type amplification, *HER2/CRYBA1* co-amplification/polysomy, and a suspected *HER2/CRYBA1* translocation/fusion pattern. Cluster-type amplification was characterized by closely stippled adjacent numerous signals that form a large visible cluster. Cluster-type amplification was detected in three out of eight cases (38%).

The pattern of a co-amplification or a polysomy was identified in 3 out of 112 tumors and was characterized by increased copy numbers of both *HER2* and *CRYBA1* (>4 copy number, Figure 4C), one of which *HER2*-amplified. The co-localization pattern was recognized in 20 out of 112 carcinomas (23%). An atypical pattern characterized by a close co-localization of *HER2* and *CRYBA1* was interpreted as translocation and fusion of the two genes (Figure 4D). 

By comparing the *HER2* gene copy number and immunohistochemical protein expression, the amplified cases were both immunohistochemically positive 3+ (2/8; 25%), equivocal 2+ (3/8; 38%), and negative 0 (1/8; 13%) and 1+ (2/8; 25%).

Tumors not amplified were both immunohistochemically negative 0 (10/82; 12%) and 1+ (32/82; 39%), equivocal 2+ (28/82; 34%), and positive 3+ (12/82; 15%). 

A subset of the *HER2*-amplified cases shows a co-expression of ER (HER2+/ER+ in three out of eight tumors). 

HER2 overexpression is significantly correlated with an increased number of *HER2* gene copies (*p* = 0.01, R = 0.24; Spearman test; Figure 4E) but not with *HER2* amplification status based on the human ASCO/CAP score for both single (*p* = 0.88, R = 0.01; Spearman test) and double-probe evaluation system (*p* = 0.84, R = 0.02; Spearman test).

The *HER2* amplification status was not correlated with tumor size (*p* = 0.88, R = 0.05; Spearman test), histological grade (*p* = 0.4, R = 0.09; Spearman test), lymph node metastasis (*p* = 0.39 R−0.13 Spearman test), and systemic metastasis (*p* = 0.21, R = 0.23; Spearman test). The OS and tumor-specific survival cannot be determined due to a small sample size of the *HER2* amplified cases. Statistically significant results are summarized in Table 1, and overall data are presented in Appendix A. 

## 4. Discussion

HER2 is an important oncogenic driver in 10–20% of invasive breast cancer [2], and its amplification is highly correlated with protein overexpression [3]. Targeted therapies, especially considering the HER2-positive subtype, have improved the prognosis of breast cancer. A total of 15–20% of patients with early breast cancer exhibit HER2 receptor overexpression, gene amplification, or both, and currently, the standard of care for these women is trastuzumab, a monoclonal antibody targeting the extracellular regions of the HER-2 tyrosine kinase receptor. This significantly improves OS and disease-free survival in women with HER2-positive early breast cancer and HER2-positive metastatic breast cancer [34,35]. Different clinical trials have shown the good therapeutic potential of other anti-HER2 agents, such as pertuzumab (a monoclonal antibody), lapatinib, and afatinib (tyrosine kinase inhibitor molecules) [35,36].

HER2 is significantly overexpressed in a variety of canine tumors. In canine anal sac gland carcinoma, HER2 mRNA and protein expression are significantly higher than in normal anal sac tissue [19]. The HER2 protein is overexpressed in a subset of canine urothelial carcinoma compared with the normal urothelium, but no association with the clinical parameters has been found [20]. In canine transitional cell carcinoma, lapatinib inhibits the phosphorylation of HER2 and cell tumor growth in vitro and reduces the tumor volume in vivo, suggesting that the molecular target also exerts its role in dogs, inhibiting HER2 signaling and inducing cell cycle arrest [37].

The significance of HER2 in canine mammary carcinoma is highly controversial, and several studies have analyzed this topic [10,11,12,13,14,15,16]. A multiplexed branched-DNA assay revealed that *HER2* gene expression is increased in benign and malignant canine mammary tumors compared with normal healthy mammary glands [38], and *HER2*, *HER3*, and *HER4* are amplified in canine mammary tumor cell lines [39]. HER2 protein expression has been investigated in canine mammary carcinomas, and its prevalence is very heterogeneous based on different studies, evaluation methods, and antibodies used. Abadie et al. [40] found no HER2-positive cases in 350 canine mammary carcinomas. Conversely, other studies found that the percentage of HER2-positive carcinomas ranged from 15% to 35.3% [13,16,41,42].

The biological meaning and the method of detecting HER2 in canine mammary carcinoma are objects of debate. The specificity of commercial anti-human HER2 antibody (Dako A0485) was raised by Burrai and colleagues [43]. Using Western immunoblot, the authors found bands at unexpected molecular weights in dogs, interpreted as a lack of cross-reactivity to the commercial antibodies available and produced against human HER2. The results were also corroborated by a lack of specificity with reverse-phase protein arrays. Later, Tsuobi et al., in 2019, demonstrated commercial polyclonal antibody specificity on canine urothelial carcinoma cell lines, finding a single band at 185 kDa [20]. To validate the specificity of HER2 antibody in our samples, immunoblot anti-HER2 was performed on protein lysate from canine mammary carcinoma cell lines, and we demonstrated that this antibody specifically recognizes canine HER2 corresponding to 185KD in a human control. These data corroborate the specificity of this antibody in canine species. 

In accordance with what has already been discussed by Peña and colleagues, the differences in the literature regarding HER2 in canine mammary tumors can be attributed to technical and reaction assessment variables [30]. Pre-analytical, analytical, and post-analytical factors should be addressed [30]. Common errors in the pre-analytical phase of immunohistochemistry are overfixation or delayed fixation [44,45]. Cold ischemia, also called ischemic time, is described [46], and specifically, a delay of fixation of more than 2 h decreases the reactivity to ER, PR, and HER2 in human breast cancer [46]. Therefore, rigorous methodological and evaluation criteria should be followed to standardize the results in veterinary medicine and obtain comparable data in the literature.

The prognostic role of HER2 is controversial in dogs, as several studies conducted on canine mammary carcinoma have shown contrasting results. In our series, there is an overexpression of HER2 that is not associated with other clinical–pathological parameters, such as tumor size, histological grade, Ki67 index, and ER expression. In the literature, HER2 is considered variably correlated to aggressive biological behavior of the neoplasm, such as histological grade, nuclear pleomorphism, mitotic count [14,15], and poor survival [14,24,47]. Moreover, the pathological features of increased malignancy, such as tumor invasion, were associated with HER2, demonstrated by significant EGFR and HER2 mRNA expression in invasive matrix-producing carcinoma compared to in situ carcinoma [48]. Conversely, other studies indicate an association of HER2 with better survival [13], and others find no association with grade [12], prognosis [41], or prognostic variables [11,49]. On the other hand, regarding the other tested markers, in our caseload, in line with the literature [28], a higher Ki67 index correlated with poor biological behavior and increased metastatic potential, while ER-expressing tumors showed associations with lower histological grade, tumor size, and Ki67 index. 

The amplification status of *HER2* was already investigated in a small number of canine mammary carcinoma via chromogenic in situ hybridization by De Las Mulas et al. [24], and no *HER2* amplification was found. In the present study, we performed single- and dual-probe fluorescent in situ hybridization, and 10% of the carcinomas showed amplification of *HER2*. We used two different FISH approaches to assess chromosomal abnormalities, co-amplification or polysomy and translocations, which were identified with dual-probe FISH and not single-probe FISH. HER2 overexpression was significantly correlated with increased numbers of *HER2* gene copies but not with the amplification status based on the ASCO/CAP evaluation system. This indicates that overexpression of the protein is parallel to the copy number variations, but the evaluation system built around the prognostic and therapeutic data of human breast cancer is not adaptable, as it is used for canine mammary carcinoma. 

In addition to the trend of HER2 overexpression and amplification, in our subset of tumors, we found co-amplification and polysomy (3/112; 3%) and a subgroup of tumors negative to immunohistochemistry but positive to *HER2* gene amplification (3/8; 37.5%); this lack of concordance between immunohistochemistry and ISH has recently been described in human breast cancer [50]. This condition as it occurred in our caseload was suggested to be caused by post-transcriptional mechanisms [50]. Despite the relative absence of detectable protein among these cases, the patients showed partial or complete clinical response to neoadjuvant combined therapy, including a HER2 blocking agent, suggesting the importance of combing testing of both IHC and ISH [50].

Remarkably, we observed a signal of possible *HER2/CRYBA1* translocation in 23% of the caseload, which had never been previously reported in the human or veterinary literature. The pattern is the same as that found with break-apart probes, indicating a possible translocation of the *HER2* oncogene in proximity to the *CRYBA1* gene since the two are normally distant from each other. *HER2* is found at 22.760373-22.785367K and *CRYBA1* at 43.372507-43.378561K on canine chromosome 9. This distance implies two distinct, non-close FISH signals, as seen in 77% of the carcinomas analyzed. Further investigations with next-generation sequencing are required to confirm these data, and the mechanism of loss of heterozygosity should be investigated (LOH). LOH is a genetic event characterized by the loss of one allele that frequently causes the inactivation of tumor-suppressor genes in human carcinogenesis [51]. The underlying mechanism of LOH concerns structural chromosomal alterations, of which the most frequently described is an unbalanced translocation; therefore, although LOH has been described more commonly in tumor-suppressor genes, this genetic mechanism must be considered in the present caseload [51].

Based on these preliminary findings, HER2 does not seem to be associated with robust clinical–pathological malignancy parameters in canine mammary tumors, although the small number of HER2-positive cases can lead to an underestimation of its biological role, and it should be complemented with further prognostic studies. 

It is believed that gene amplification, defined as an increase in the DNA copy number, is a major molecular mechanism driving oncogenesis in many kinds of cancer, promoting tumor progression [52,53]. DNA amplification is a major source of genetic variation, which is not necessarily always pathogenic, but can lead to genetic polymorphisms in humans and animals [52,54]. The underlying mechanism of gene amplification is largely unknown; canonical theory concerns the breakage–fusion–bridge cycle, characterized by the repeated breakage and fusion of isochromatids after telomere loss and resulting inverted duplication. Recently a mechanism behind small DNA fragments was proposed that small DNA molecules in the form of synthetic oligonucleotides can be considered potent tools for genomic rearrangement regardless of the presence of double-strand breakage in the target DNA [52].

In our caseload, we found a pattern of amplification, co-amplification, and break-apart expression, which implies an underlying genetic alteration mechanism for which *HER2* can be a passenger bystander gene.

## 5. Conclusions

In conclusion, the *HER2* gene was found to be amplified in a subgroup of canine mammary carcinomas, and the *HER2* gene copy number correlated with HER2 overexpression but not with tumor biological behavior. Surprisingly, a possible translocation of *HER2/CRYBA1* was found that had never been described in the literature, although it remains to be confirmed on next-generation sequencing.

## Figures and Tables

**Figure 1 vetsci-09-00583-f001:**
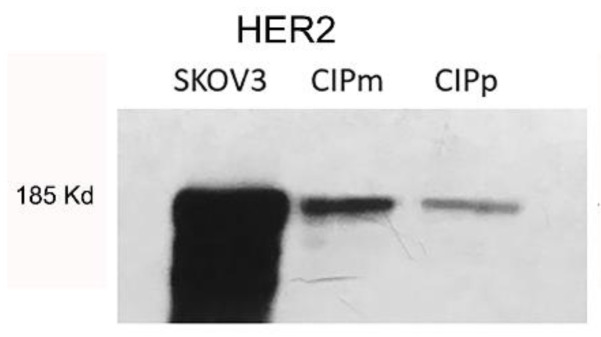
Specific bands corresponding to 185KD HER2 are present in canine cell lines (CYPp and CYPm) co-migrating with human HER2 over-expressed in human SKOV3 cell line. (Original figure: Appendix A).

**Figure 2 vetsci-09-00583-f002:**
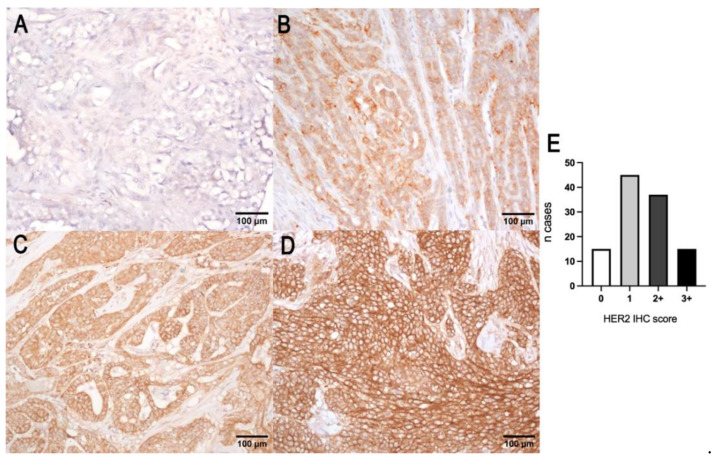
HER2 protein score of canine mammary tumors. (**A**) Negative (0), showing no immuno-reactivity for HER2; (**B**) negative (1+), characterized by a weak incomplete membrane expression of HER2; (**C**) equivocal (2+), showing a complete or basolateral moderate HER2 expression; (**D**) overexpression (3+), characterized by an intense complete membranous labeling; (**E**) histogram representation of the number of cases showing the 4 scores of the IHC of HER2.

**Figure 3 vetsci-09-00583-f003:**
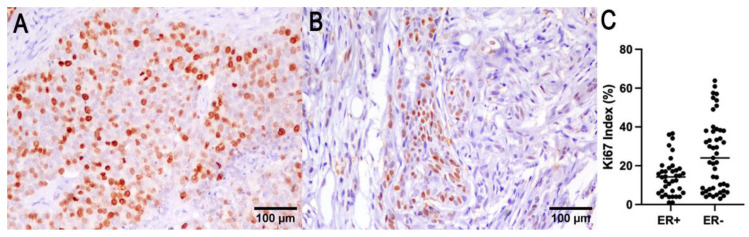
Ki67 (**A**) and ER (**B**) immunohistochemistry of a representative CMC. (**C**) ER expression was found to be correlated with a lower Ki67 index.

**Table 1 vetsci-09-00583-t001:** Statistically significant association between clinical–pathological parameters, HER2 protein, HER2 gene, ER, and Ki67 index. HER2: human epidermal growth factor receptor 2; ER: estrogen receptor; OS: overall survival.

	Histological Grade	Tumor Size	Lymph Node Metastases	Systemic Metastases	OS	HER2 Protein Expression	Ki67 Index	ER exPression	*HER2* Gene Copies
Histological grade	-	*p* = 0.01 R = 0.37	*p* < 0.0001 R = 0.64	*p* = 0.004 R = 0.50	*p* = 0.01	ns	ns	*p* = 0.003 R = −0.31	ns
Tumor size	*p* = 0.01 R = 0.372	*-*	ns	ns	*p* = 0.01 R = −039	ns	*p* = 0.006 R = 0.37	*p* = 0.009 R−0.365	ns
Lymph node metastases	*p* < 0.0001 R = 0.64	ns	-	ns	ns	ns	*p* < 0.0001 R = 0.67	ns	ns
Systemic metastases	*p* = 0.004 R = 0.50	ns	ns	-	ns	ns	*p* = 0.006 R = 0.53	ns	ns
OS	*p* = 0.01	*p* = 0.01 R = −039	ns	ns	-	ns	ns	*p* = 0.04	ns
HER2 protein expression	ns	ns	ns	ns	ns	-	ns	ns	*p* = 0.01 R = 0.24
Ki67 index	ns	*p* = 0.006 R = 0.37	*p* < 0.0001 R = 0.67	*p* = 0.006 R = 0.53	ns	ns	*-*	*p* = 0.01 R = −0.28	Ns
ER expression	*p* = 0.003 R = −0.31	*p* = 0.009 R−0.36	ns	ns	*p* = 0.04	ns	*p* = 0.01 R = −0.28	-	ns
*HER2* gene copies	ns	ns	ns	ns	ns	*p* = 0.01 R = 0.24	ns	ns	-

## Data Availability

The data presented in this study are available on request from the corresponding author.

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
