# Peer review of "HER2 Overexpression and Cytogenetical Patterns in Canine Mammary Carcinomas"

_vetsci, 2022, doi:10.3390/vetsci9110583_

Round 1

Reviewer 1 Report

This paper is very interesting and important. Only minor details must be addressed in my opinion to improve even more this work. 

Please rephrase paragraph 52-55 to be a more fluid and understandable sentence.

"caseload" - introduce information about the number of female dogs in the study (92).

Did you use immuno for tumor classification? Please clarify. 

Your have tumors with low number such as anaplastic or ductal carcinoma among others...how did the authors perform statistical approach?  

It is important that authors must explain tumour criteria selection to apply the OS, since that they have apparently multiple tumours in female dogs. Which malignant tumour did you chose for statistical purposes?

There is no information about follow-up with exception of the time of 24 months, but how this data was collected? Did you have information about thoracic radiographs, echographs and intervals these were taken, etc…please include this information. How the authors confirm the cause of death? … there is no information about necropsy number to confirm the death cause. Did you have this information?

The authors point two different systems concerning grading malignant tumours. There have specific differences. Which the authors used effectively?

TMA were built only for benign tissue? Please clarify.

Author Response

R1: This paper is very interesting and important. Only minor details must be addressed in my opinion to improve even more this work.  

Authors: We wish to thank this reviewer for his/her kind words of appreciations and for the constructive comments. His/her comments were addressed as specified below.    

R1: Please rephrase paragraph 52-55 to be a more fluid and understandable sentence. 

Authors: The paragraph was rephrased as follow: “During the years, several monoclonal antibodies (trastuzumab, pertuzumab, lapatinib, neratinib and trastuzumab emtansine) have been developed and their therapeutic efficacy tested and validated” (ln. 53-57).  

R1: "caseload" - introduce information about the number of female dogs in the study (92). 

Authors: We added in ln. 103 and 104 the information about the sex for clarification. 

R1: Did you use immuno for tumor classification? Please clarify.  

Authors: No, we didn’t use immunohistochemistry for tumor classification; the diagnosis were based on histological criteria.  

R1: Your have tumors with low number such as anaplastic or ductal carcinoma among others...how did the authors perform statistical approach?   

Authors: Histotypes with a low number of cases were excluded from the statistical evaluation. 

R1: It is important that authors must explain tumour criteria selection to apply the OS, since that they have apparently multiple tumours in female dogs. Which malignant tumour did you chose for statistical purposes? 

Authors: Thanks for the valuable suggestion. We mistakenly included all the tumors tested, but in accordance with your suggestion we selected one tumor per patient in the case of multiple tumors and redone the survival statistic. The tumor we selected from the multiple tumors was that with highest grade. Therefore, we have added this selection criterion in the materials and methods and corrected the results. 

R1: There is no information about follow-up with exception of the time of 24 months, but how this data was collected? Did you have information about thoracic radiographs, echographs and intervals these were taken, etc…please include this information. How the authors confirm the cause of death? … there is no information about necropsy number to confirm the death cause. Did you have this information? 

Authors: Lymph node metastasis were detected by histological examination, but follow-up data of tumor recurrence, systemic metastases and survival status were collected by phone calls to the referent veterinarian or to the owner of the dogs. Unfortunately, we do not have detailed information regarding the x-rays, ultrasounds and the intervals at which they were taken. We rephrase the sentence as follows (ln. 184-187):  

“Lymph node metastasis were detected by histological examination. Tumor recurrence, systemic metastases and survival status were investigated and collected by phone calls to the referent veterinarian or to the owner of the dogs.” 

R1: The authors point two different systems concerning grading malignant tumours. There have specific differences. Which the authors used effectively? 

Authors: Canine mammary tumor grading by Pena and colleagues is based on the human breast cancer grading system by Elston and Ellis. The two grades are identical, but we have adopted that of the dog. Therefore, we have removed the human reference to avoid misunderstanding “Elston, C.W.; Ellis, I.O. Pathological Prognostic Factors in Breast Cancer. I. The Value of Histological Grade  in Breast Cancer: Experience from a Large Study with Long-Term Follow-Up. Histopathology 1991, 19, 403–410, doi:10.1111/j.1365-2559.1991.tb00229.x.” citation. 

R1: TMA were built only for benign tissue? Please clarify. 

Authors: We added in ln. 145: “eighteen blocks were composed of 110 malignant tumors” for clarification. 

Reviewer 2 Report

L.V. Muscatello et al. analyzed HER2 expression in 112 canine mammary carcinoma in the manuscript and found HER2 expression is not correlated with grade, metastases and tumor-specific survival, which is different from human mammary carcinoma. This is an important finding which shed light on species difference regarding molecular mechanisms of mammary carcinoma However, I have following concerns or suggestions for authors to consider to address, the biggest concern of which is about data presentation:

* Three antibodies HER2, Ki67 and ER were used for IHC. However, only HER2 images are shown in the manuscript (Figure 1). It is very necessary to show Ki67 and ER images to prove the accuracy of results described in text.

* For HER2 expression, showing representative IHC images with different scores would give better description of heterogeneity of HER2 expression in cases. Please also include equivocal and negative images.

* To better present the data, graphs are necessary to be included in figures together with IHC or in situ images. For instance, add histograms of quantification of samples with different scores based on HER2 IHC. More graphs are needed for Ki67 results and ER results as well.

* Please add scale bars in IHC images.

* In figure2, it would be better if authors can add more images with lower magnification to show the morphology of the whole tissue analyzed in the study.

* Line 38. It's better to be cautious to draw "functional" conclusion from expression results only.  "HER2 does not seem to play a driver role" may sound misleading to readers.

*Line 228. Be consistent when you use numbers or words in text. "37 of 117" or " Thirty-seven out of one hundred and twelve"

*Line 283. "morphological translocation" may not be the appropriate word to describe the results

Author Response

R2: L.V. Muscatello et al. analyzed HER2 expression in 112 canine mammary carcinoma in the manuscript and found HER2 expression is not correlated with grade, metastases and tumor-specific survival, which is different from human mammary carcinoma. This is an important finding which shed light on species difference regarding molecular mechanisms of mammary carcinoma However, I have following concerns or suggestions for authors to consider to address, the biggest concern of which is about data presentation: 

Authors: We wish to thank this reviewer for her/his kind word of appreciation of our work, for the careful review of all the data and for her/his constructive comments that have greatly increased the precision of the presentation of the data. His/her comments were addressed as specified below. 

R2: * Three antibodies HER2, Ki67 and ER were used for IHC. However, only HER2 images are shown in the manuscript (Figure 1). It is very necessary to show Ki67 and ER images to prove the accuracy of results described in text. 

Authors: Figures depicting Ki67 and ER have been added (Figure 3).  

R2: * For HER2 expression, showing representative IHC images with different scores would give better description of heterogeneity of HER2 expression in cases. Please also include equivocal and negative images. 

Authors: The figures of the different HER2 score have been added (Figure 2).  

R2: * To better present the data, graphs are necessary to be included in figures together with IHC or in situ images. For instance, add histograms of quantification of samples with different scores based on HER2 IHC. More graphs are needed for Ki67 results and ER results as well. 

Authors: The graphs have been included (Fig 2E, Fig 3C, Fig 4E).  

R2: * Please add scale bars in IHC images. 

Authors: The scale bars have been added.  

R2: * In figure2, it would be better if authors can add more images with lower magnification to show the morphology of the whole tissue analyzed in the study. 

Authors: In accordance with what is presented in human literature for fluorescence in situ hybridization (Press et al. HER2 Gene Amplification Testing by Fluorescent In Situ Hybridization (FISH): Comparison of the ASCO-College of American Pathologists Guidelines With FISH Scores Used for Enrollment in Breast Cancer International Research Group Clinical Trials. We include a small magnification figure here to show homogeneous expression in the nuclei. J Clin Oncol. 2016 Oct 10;34(29):3518-3528.doi: 10.1200/JCO.2016.66.6693.), the images are always presented at high magnification with a large close-up on the nuclei, as it would be difficult to see the signals with a lower magnification. A lower magnification figure is included below to show homogeneous expression in the nuclei in a co-amplified case for HER2 (red) and CRYBA1 (green). 

R2: * Line 38. It's better to be cautious to draw "functional" conclusion from expression results only.  "HER2 does not seem to play a driver role" may sound misleading to readers. 

Authors: Lines 38-39 and 435-439 we replace the term driver with the following sentence:  

“HER2 is not associate with clinical-pathological parameters of malignancy in canine mammary tumors”.  

R2: *Line 228. Be consistent when you use numbers or words in text. "37 of 117" or " Thirty-seven out of one hundred and twelve" 

Authors: We replaced “112” with “one hundred and twelve”. 

R2: *Line 283. "morphological translocation" may not be the appropriate word to describe the results 

Authors: We removed the term “morphological”. 

Reviewer 3 Report

In the MS entitled “HER2 overexpression and cytogenetical patterns in canine mammary carcinomas” by L.V. Muscatello and collaborators the Authors address the incidence of HER2 gene amplification in canine mammary carcinomas (CMCs; n=112) by FISH methods, its correlation with increased Her2 protein expression (by immunohistochemistry/IHC) and with variables currently related to clinical prognosis (histological grade, tumor size, metastases) and, finally, the technical (but relevant) issue of the specificity of antibodies used to detect Her2. By certified criteria only ~10% (n=8) of the tumors that revealed as appropriate for full analysis were shown to overexpress Her2. There was some correlation between overexpression of Her2 (by IHC) and reduced overall survival. The Authors, however, conclude on the unlikely relevant role of HER2 as a driver oncogene in CMCs, although they recognize the need of further research to strengthen such conclusion. This would be in stark contrast with the evidence obtained from Her2+ breast cancers in humans in which case the introduction of a monoclonal antibody targeting Her2 (trastuzumab) drastically changed clinical outcomes; this was subsequently corroborated by the introduction of additional highly effective antibodies (and small molecule inhibitors) targeting Her2.

This research looks mostly carefully performed, and the efforts to provide appropriate controls are notorious, although not free of criticism. The major caveat with this research is the low number of Her2+ cases (n=15) precluding the formulation of more robust conclusions on the issue

SPECIFIC REMARKS:

1. Regarding the antibody used to detect Her2 by IHC and its characterization: A) why the Authors utilized a polyclonal antibody? Typically, monoclonals, given their potential for unlimited production over time are more suitable for future reproducibility of data; B) in the characterization of the Her2 -specific antibody used herein the Authors utilized a certified human ovarian cell line known to overexpress Her2 (SKOV3) and 2 canine cell lines; a more appropriate set of controls would be a human cell line overexpressing Her2 (eg, SK-BR3, or SKOV3), a human cell line NOT expressing Her2 (eg, MDA-MB 231), and 1 to 2 canine cell lines expressing Her2, PLUS extracts of the cell line overexpressing Her2 and the canine lines depleted for Her2 (eg, siRNA technology). Also, the band corresponding to Her2 in the overexpressing cell line (Fig.1) looks full of faster migrating species, possibly corresponding to degradation products! Finally, loading controls shall be provided (Ponceau staining of the membranes, plus immunoblotting for, eg, beta-actin, or tubulin, or a histone); Loading controls are MANDATORY!

2. Regarding the FISH data in Fig. 2D, if a fusion (via translocation) between HER2 and CRYBA1 has indeed occurred, as the images suggest (overlapped green-red signal), where can the signals of the non-fused alleles be found - ie, independent red-only and green-only signals??? (Please, cf the exemplary case of the classical PML-RARA translocation as visualised by interphase FISH); those independent signals are not apparent in the shown image. The alternative explanation would be a loss-of-heterozygosity/LOH following the putative HER2/CRYBA1 translocation, resulting in only-fusion (overlapped) signals; LOH, although common for onco-suppressors is however atypical for oncogenes. The Authors shall comment on this issue.

3. Although the individual microscopy images show-up in good quality the overall aesthetic appearance of the corresponding Figures looks terrible; the Authors shall improve this.

4. The data on the expression of markers used herein (Her2, Ki-67, ER/Estrogen receptor) and the data on gene amplification (single, dual evaluations), plus the corresponding statistics should be compiled in a Table (or else 2 tables) to facilitate a comprehensive overview of the data.

5. The style of the English language, as well as text clarity and flow of information shall be improved; typos (frequent) shall be eliminated.

6. In Discussion some topics, such as the issue of the breakage-bridge-fusion cycle (which relates more to telomere attrition, indeed) should be more professionally addressed and contextualized.

7. Make clearer the reason for using two probes for FISH to conclude on, or exclude, polysomy; this will be important for the non-expert reader

Author Response

R3: In the MS entitled “HER2 overexpression and cytogenetical patterns in canine mammary carcinomas” by L.V. Muscatello and collaborators the Authors address the incidence of HER2 gene amplification in canine mammary carcinomas (CMCs; n=112) by FISH methods, its correlation with increased Her2 protein expression (by immunohistochemistry/IHC) and with variables currently related to clinical prognosis (histological grade, tumor size, metastases) and, finally, the technical (but relevant) issue of the specificity of antibodies used to detect Her2. By certified criteria only ~10% (n=8) of the tumors that revealed as appropriate for full analysis were shown to overexpress Her2. There was some correlation between overexpression of Her2 (by IHC) and reduced overall survival. The Authors, however, conclude on the unlikely relevant role of HER2 as a driver oncogene in CMCs, although they recognize the need of further research to strengthen such conclusion. This would be in stark contrast with the evidence obtained from Her2+ breast cancers in humans in which case the introduction of a monoclonal antibody targeting Her2 (trastuzumab) drastically changed clinical outcomes; this was subsequently corroborated by the introduction of additional highly effective antibodies (and small molecule inhibitors) targeting Her2. 

R3: This research looks mostly carefully performed, and the efforts to provide appropriate controls are notorious, although not free of criticism. The major caveat with this research is the low number of Her2+ cases (n=15) precluding the formulation of more robust conclusions on the issue. 

Authors: Dear Reviewer,  

thank you for the careful review, for the criticisms and suggestions that will surely lead to an improvement of the manuscript. The percentage of the HER2 positive cases is comparable to that described in human literature and the total number of tumours tested is on average high, compared to the works in the literature in canine mammary tumours, also considering the high cost of the fish method. We agree that the small number of HER2 positive cases is certainly important in the interpretation of the work; therefore, we have added this limitation in the discussion (lines 435-439) and as follows: “Based on these preliminary findings, HER2 doesn’t seem to be associated with robust clinical-pathological parameters of malignancy in canine mammary tumors although the small number of HER2 positive cases can lead to an underestimate in the interpretation of its biological role and it should be complemented with further prognostic studies”.  

SPECIFIC REMARKS: 

R3: 1. Regarding the antibody used to detect Her2 by IHC and its characterization: A) why the Authors utilized a polyclonal antibody? Typically, monoclonals, given their potential for unlimited production over time are more suitable for future reproducibility of data; B) in the characterization of the Her2 -specific antibody used herein the Authors utilized a certified human ovarian cell line known to overexpress Her2 (SKOV3) and 2 canine cell lines; a more appropriate set of controls would be a human cell line overexpressing Her2 (eg, SK-BR3, or SKOV3), a human cell line NOT expressing Her2 (eg, MDA-MB 231), and 1 to 2 canine cell lines expressing Her2, PLUS extracts of the cell line overexpressing Her2 and the canine lines depleted for Her2 (eg, siRNA technology). Also, the band corresponding to Her2 in the overexpressing cell line (Fig.1) looks full of faster migrating species, possibly corresponding to degradation products! Finally, loading controls shall be provided (Ponceau staining of the membranes, plus immunoblotting for, eg, beta-actin, or tubulin, or a histone); Loading controls are MANDATORY! 

Authors: We thank the reviewers for these specific considerations regarding the specificity of HER2 antibody used in this paper. We will try to answer point by point: 

  1. A) We are agreed with the reviewer that generally monoclonal antibodies are strongly more specific than polyclonal one. Otherwise when we use monoclonal antibody in different species, usually we can have less sensitivity and specificity. HER2 polyclonal antibody used in this paper (DAKO) is one of the oldest and largely used in human research (Mod Pathol. 2002 Jun;15(6):657-65.) and has been also tested in feline and canine species (Vet Pathol. 2019 May;56(3):369-376.; Microscopy and Microanalysis, 19(4), 876-882.). Additionally, as shown below monoclonal antibody used on the same samples gave a less specificity and also the quantity of the proteins detected was lower. That has been the rationale to use these polyclonal antibody in this paper.

In addition, we did not have good results with preliminary tests on tumor sections in immunohistochemistry. Considering that the Dako HER2 A0485 polyclonal had already been validated in the canine urinary bladder by Tsuobi and collegues (Vet Pathol. 2019 May;56(3):369-376. doi: 10.1177/0300985818817024.  Epub 2019 Jan 6) (as per blot below) we decided to use this antibody.  

  1. B) We used SKOV3 overexpressing human cell line as positive control and two canine cell lines. We retain to do not use further negative controls as well to apply shRNA technology to validate the antibody specificity because we found a specific band co-migrating with human HER2 of 185kD. Additionally, in literature these antibodies gave the same results also in western blot. Regarding the presence of a possible denaturated proteins in SKOV3 lane, we retain that the smearing effect is due to the high amount of proteins loaded and because SKOV3 is a HER2 overexpressing cell line; these excesses gave a smearing effect in the western blot. Regarding the loading control in the following figure the alpha tubulin is shown demonstrating the same quantity of proteins loaded in the three samples. 

R3: 2. Regarding the FISH data in Fig. 2D, if a fusion (via translocation) between HER2 and CRYBA1 has indeed occurred, as the images suggest (overlapped green-red signal), where can the signals of the non-fused alleles be found - ie, independent red-only and green-only signals??? cPlease, cf the exemplary case of the classical PML-RARA translocation as visualised by interphase FISH); those independent signals are not apparent in the shown image. The alternative explanation would be a loss-of-heterozygosity/LOH following the putative HER2/CRYBA1 translocation, resulting in only-fusion (overlapped) signals; LOH, although common for onco-suppressors is however atypical for oncogenes. The Authors shall comment on this issue. 

Authors: Thanks for the interesting comment. We have included a new photo that also shows the unfused alleles. As suggested, we believe that we found a fusion signal considering that LOH occurs more commonly for tumor suppressor genes; we have added a comment on this topic in the discussion (ln 427-434) and as follows:  

“Further investigations with the Next Generation Sequencing methods are required to confirm these data and the mechanism of loss of heterozygosity should be investigated (LOH). LOH is a genetic event characterized by the loss of one allele that frequently causes inactivation of tumor suppressor genes in human carcinogenesis. The underlying mechanism of LOH concerns structural chromosomal alterations, of which the most frequent described is an unbalanced translocation, therefore, although LOH has been described more commonly in tumor suppressor genes, it is necessary that this genetic mechanism be considered in the present caseload”.  

R3: 3. Although the individual microscopy images show-up in good quality the overall aesthetic appearance of the corresponding Figures looks terrible; the Authors shall improve this. 

Authors: The figures plate has been updated with the addition of a figure D showing the unfused alleles in addition to the fused ones. In addition, we have inserted a chart as requested by reviewer 2. 

R3: 4. The data on the expression of markers used herein (Her2, Ki-67, ER/Estrogen receptor) and the data on gene amplification (single, dual evaluations), plus the corresponding statistics should be compiled in a Table (or else 2 tables) to facilitate a comprehensive overview of the data. 

Authors: We added a table 1 that summarize the statistically significant association and a supplementary table 2, which contains all the data.  

R3: 5. The style of the English language, as well as text clarity and flow of information shall be improved; typos (frequent) shall be eliminated. 

Authors: The English language have been improved. 

R3: 6. In Discussion some topics, such as the issue of the breakage-bridge-fusion cycle (which relates more to telomere attrition, indeed) should be more professionally addressed and contextualized. 

Authors: we added a paragraph regarding these topics (ln 446-449).  

R3: 7. Make clearer the reason for using two probes for FISH to conclude on, or exclude, polysomy; this will be important for the non-expert reader. 

Authors: We would like to thanks the reviewer for the constructive comment. We add in the discussion a sentence highlighting this point. “We used the two different FISH approach in order to assess chromosomal abnormalities, including co-amplification or polysomy and translocations, thus identified with dual-probe FISH and not with a single probe FISH” ln. 403-405. 

Round 2

Reviewer 2 Report

The current version is substantially improved and meet the standard and requirement for publication

Author Response

Thank you for your revision

Reviewer 3 Report

General Comment: The MS has improved considerably

MINOR CORRECTIONS

1) (Line 38), in ABSTRACT:… “”Her 2 is not associated with clinical-pathological parameters of malignancy in canine mammary tumors …”” ----Should be:  “”HER2 is not associated with clinical-pathological parameters of increased malignancy in canine mammary tumors “” // Clearly, these tumors are malignant!! A similar correction shall be done in DISCUSSION, Lines 431-432

2) (Figure 4D) FISH HER2/CRYBA1 fusion pattern; arrows, and arrowheads should highlight fusion and singlet signals, for clarity of interpretation.

3) English style still has room for some improvement; the Authors should seek for some help (or the Journal may provide such service). The comment by Authors in this regard is telling (transcribed below; it contains a grammatical error). Please, take this criticism as constructive!

Authors: The English language have been improved.

Author Response

R3: General Comment: The MS has improved considerably

MINOR CORRECTIONS

R3: 1) (Line 38), in ABSTRACT:… “”Her 2 is not associated with clinical-pathological parameters of malignancy in canine mammary tumors …”” ----Should be:  “”HER2 is not associated with clinical-pathological parameters of increased malignancy in canine mammary tumors “” // Clearly, these tumors are malignant!! A similar correction shall be done in DISCUSSION, Lines 431-432

Authors: These sentences have been modified as suggested.

R3: 2) (Figure 4D) FISH HER2/CRYBA1 fusion pattern; arrows, and arrowheads should highlight fusion and singlet signals, for clarity of interpretation.

Authors: The arrows and arrowheads have been added to Figure 4.

R3: 3English style still has room for some improvement; the Authors should seek for some help (or the Journal may provide such service). The comment by Authors in this regard is telling (transcribed below; it contains a grammatical error). Please, take this criticism as constructive!

Authors: The English language have been improved.

Authors: The manuscript has been revised by an English native speaker.